



# Community-driven natural hazard and physical vulnerability assessment in a disaster-prone urban neighborhood

Alejandro Builes-Jaramillo[1], Annie E.G. Winson[2], Emma Bee[2], Nancy Quirós[3], Dairo Urán[3], James Rúa[3], Luis Alejandro Rivera-Flórez[4,5,1], Camilo Restrepo-Estrada[5], Ingry Natalia Gómez-Miranda[6], Claire Dashwood[2], and João Porto de Albuquerque[7]

[1]Institución Universitaria Colegio Mayor de Antioquia, Carrera 78 # 65-46 Bloque Fundacional piso 2, Medellín, Colombia.
[2]British Geological Survey. Nicker Hill, Keyworth, Nottingham, NG12 5GG, United Kingdom
[3]Junta de Acción Comunal El Pacífico. Medellín, Colombia
[4]CentroGeo, México
[5]Universidad de Antioquia, Colombia
[6]Institución Universitaria Pascual Bravo, Colombia
[7]School of Social and Political Sciences, University of Glasgow, Glasgow, United Kingdom

**Correspondence:** Alejandro Builes-Jaramillo (luis.builes@colmayor.edu.co) and Annie E.G. Winson (anwin@bgs.ac.uk)

**Abstract.** Effectively reducing the risk of disasters in urban neighbourhoods is a key policy priority, which is becoming more pressing due to climate change. However, disaster risk reduction and climate adaptation efforts are often hampered by data gaps regarding the physical vulnerability and local impacts of hazards at the neighbourhood level. These gaps are particularly pronounced for informal settlements and marginalized communities of cities in the Global South, which are frequently invisible
5 in official hazard and risk maps. Community-generated data and participatory methods are promising approaches to address these gaps, but there is a lack of guidelines and empirical examples of effective integration of communities into vulnerability assessment. This study presents the co-production of a physical vulnerability assessment framework, between academia, practitioners, and community researchers, using an iterative and easily replicable methodology. Working with community researchers from the self-constructed community El Pacífico in Medellín (Colombia), we developed a hazard perception exercise
10 based on vulnerability indicators and produced hazard perception and physical vulnerability usable maps. We show how this work was able to refine the spatial scale of the hazard maps available for the neighbourhood, going beyond the city planning tools and enabling a building-scale vulnerability assessment that is valuable not only to support community decision-making and planning but also to advocate for public interventions towards reducing disaster risks.

## 1 Introduction

15 The United Nations Agency for Disaster Risk Reduction defines vulnerability as the conditions shaped by physical, social, economic, and environmental factors or processes that heighten the susceptibility of an individual, a community, an asset, or system to the impacts of hazards (UNDRR, 2017). Fast-growing cities around the developing world show the relationship between deprivation and vulnerability (Kabiru et al., 2023; Jankowska et al., 2011), the rapid growth of urban settlements with precarious infrastructure and without access to basic services presents a major challenge to disaster risk management in





most Latin American cities. These communities tend to face higher vulnerability factors (Abunyewah et al., 2018; Williams et al., 2019) which adds to the fact that they are located within areas highly susceptible to disasters. The impact of catastrophic events triggered by natural hazards and climate change are largely determined by the settlements' exposure and vulnerability (Handmer et al., 2012; Bello et al., 2021).

Effectively reducing the risk of disasters in these urban deprived neighborhoods is a key policy priority for many coun-
tries in the global South (Ulbrich et al., 2023; UNDRR, 2022), and is becoming more pressing due to climate change (Khan et al., 2022). The effort is hampered by data gaps regarding hazard susceptibility and physical vulnerability of these urban neighborhoods which limits the establishment of effective disaster risk reduction (Sachs et al., 2020). For example, current official hazard maps from institutional disaster risk and land use planning in Medellín, Colombia, do not have a sufficiently high resolution to capture all neighborhoods and to fully represent communities' risks (Municipio de Medellín, 2014). As with
many other locations, this data inequality is strongly associated with social inequality; i.e., there is less, lower-quality data available for the most deprived and impoverished urban areas. Compounding this, these areas tend to be the locations that are most exposed to natural hazards (Porto de Albuquerque et al., 2023).

The generation of data by citizens and communities is considered an effective mechanism to fill the data and knowledge gaps regarding urban development challenges and environmental change impacts (de Sherbinin et al., 2021; Fraisl et al.,
2022). However, integrating community-generated data into methods for assessing the local vulnerability conditions and hazard impacts requires careful investigation, curation, and validation. Physical vulnerability can be assessed with fragility or damage curves (Peduto et al., 2017; Tarbotton et al., 2015), damage matrices (Menoni, 2006), or vulnerability indicators (Papathoma-Köhle et al., 2017; Papathoma-Köhle, 2016). All those methods have shortcomings when used in data-scarce contexts like the ones in the urban/rural border of Medellín where hazard information lacks detail, the interaction between the hazards and the
buildings is difficult to model, and/or there is no geolocation or geoinformation available. One approach is to use community-generated data (Sekajugo et al., 2022; Hicks et al., 2019; Ferri et al., 2020) to overcome these issues and provide detailed information for the physical vulnerability analysis. However, there is a lack of guidelines for co

To address this knowledge gap, we propose a method for enhancing the indicators approach to incorporate community goals, community information, and a feedback process where community researchers play a key role in the physical vulnerability
analysis. We introduce a co-created approach to assess the physical vulnerability of the community to three distinct hazards: landslides, floods, and rockfalls. This approach bridges the gap between current official hazard assessments, which often lack specificity and context, and the community needs through co-designing the methodology for data creation with the community.

Additionally, we show how this approach contributed to strengthening the community's capacity to identify precautionary actions for reducing and managing risks effectively. The results presented in this article are one of the outcomes of the URBE
Latam project (https://gtr.ukri.org/projects?ref=ES%2FT003294%2F1), which aimed to bridge the gap between sustainable development and equitable resilience. URBE Latam employed a transdisciplinary research approach to empower residents in disaster-prone urban poor neighborhoods.

This study focuses on work developed with the El Pacífico community, located on the urban-rural border of Medellín, Colombia (Figure 1). In this study, we worked with the community to i) increase the resolution of the existing landslide hazard



map; ii) create a community hazard assessment for flood, rockfall, and landslide; iii) perform a community building survey and code for building type, quality, etc.; iv) generate a physical vulnerability assessment for the individual hazards - landslide, flood, and rockfall. These project goals were informed by the community's objectives to understand their physical vulnerability to improve their reduction risk management strategies (Rivera Flórez et al., 2020) and, to negotiate with local authorities to keep inhabiting their socially constructed territory, which despite its risk conditions, they deem worthy of inhabiting.

In this paper, we provide a brief context about the study location (Section 2), outline the methodology that we co-created with the community to integrate perceptions of risk into hazard and vulnerability assessments alongside the final physical vulnerability maps created for each hazard (Section 3), as well as present a discussion of contributions (Section 4) followed by a short conclusion (Section 5).

## 2   Study location

El Pacífico is a self-built neighborhood located in the city of Medellín central-western urban-rural border (Fig. 1 and Fig. 2). Medellín is situated inside the narrow Aburrá Valley within the central Andes mountain range in Colombia (Figure 1). It is part of a conurbation of 10 municipalities known as the Aburrá Valley Metropolitan Area (https://www.metropol.gov.co/). El Pacífico covers an area of approximately 12,000 square meters. This community, with around 780 inhabitants living in 184 buildings is located in a high-risk area and is highly susceptible to natural hazards such as landslides and flooding due to its
location on steep slopes, downhill from areas of active deforestation (Municipio de Medellín, 2014).

Construction of the neighborhood began in the mid-1990s as a consequence of displacement caused by Colombia's armed conflict displacement. This conflict generated a migration influx into major cities, where the lack of institutional support and the inability of those displaced to return to the countryside led people to seek and build shelter on the uninhabited slopes of the city. These settlements, located outside the historical city boundaries, were constructed on steeper, more marginal land
(Comisión de la Verdad, 2022; Pérez Fonseca, 2018) making them more vulnerable to natural hazards. As a result, these areas have accounted for an overwhelming number of victims (more than 1,400 in the last 100 years due to landslides alone) and significant economic losses(Rivera Flórez et al., 2020).

By the second decade of the 21st century, the neighborhood had approximately 780 inhabitants, distributed across 184 houses (according to community leaders). Due to the continuous hazard imposed by hydro-meteorological events such as floods or
landslides, both the community and local authorities have focused on reducing the impacts of disasters in the area. Community leaders now have an advanced understanding of methods for risk reduction and risk management (Rivera Flórez et al., 2020).

### 2.1   Geological and Hydro-meteorological hazards

Due to the climatic and meteorological conditions in Medellín, areas like El Pacífico are highly susceptible to landslides, rockfalls, debris flows, and flash floods (Poveda, 2004; Poveda et al., 2006; Mejía et al., 2021; Aristizabal et al., 2022; Builes-
Jaramillo et al., 2022; Salas et al., 2022)(https://www.desinventar.net). According to the Department of Disaster Risk Management of Medellín (DAGRD, for its Spanish acronym), the city experienced more than 1,650 landslides between 1930 and

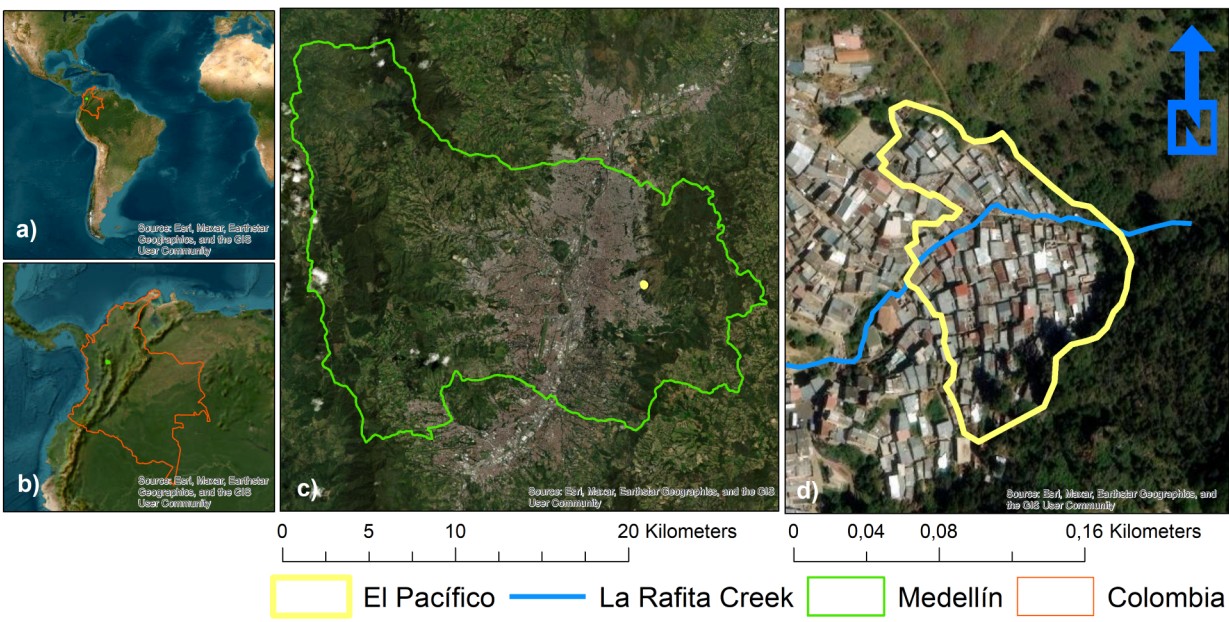

**Figure 1.** Location of El Pacífico. a) Location of Colombia and Medellín in the context of South America. b) Location of Medellín in the central Andean mountains of Colombia. c) Location of El Pacífico en the eastern slopes of Medellín. d) Location of El Pacífico in the urban-rural border of the city.

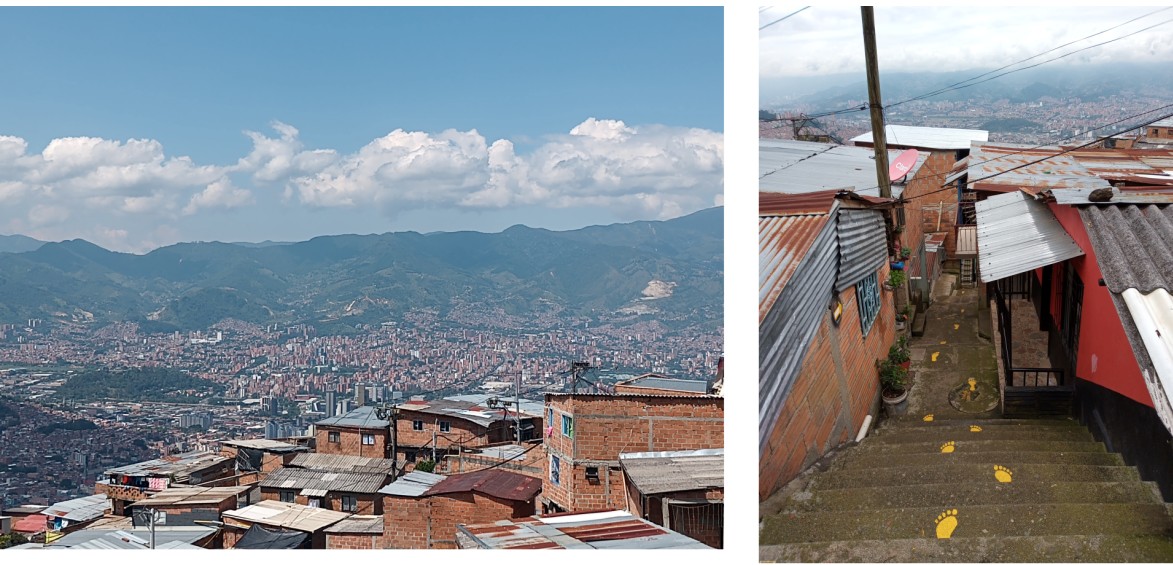

**Figure 2.** left) View of Medellín from El Pacífico. right) Stairs inside the neighborhood with yellow footprints are made as community-made evacuation routes. Photos: Alejandro Builes-Jaramillo


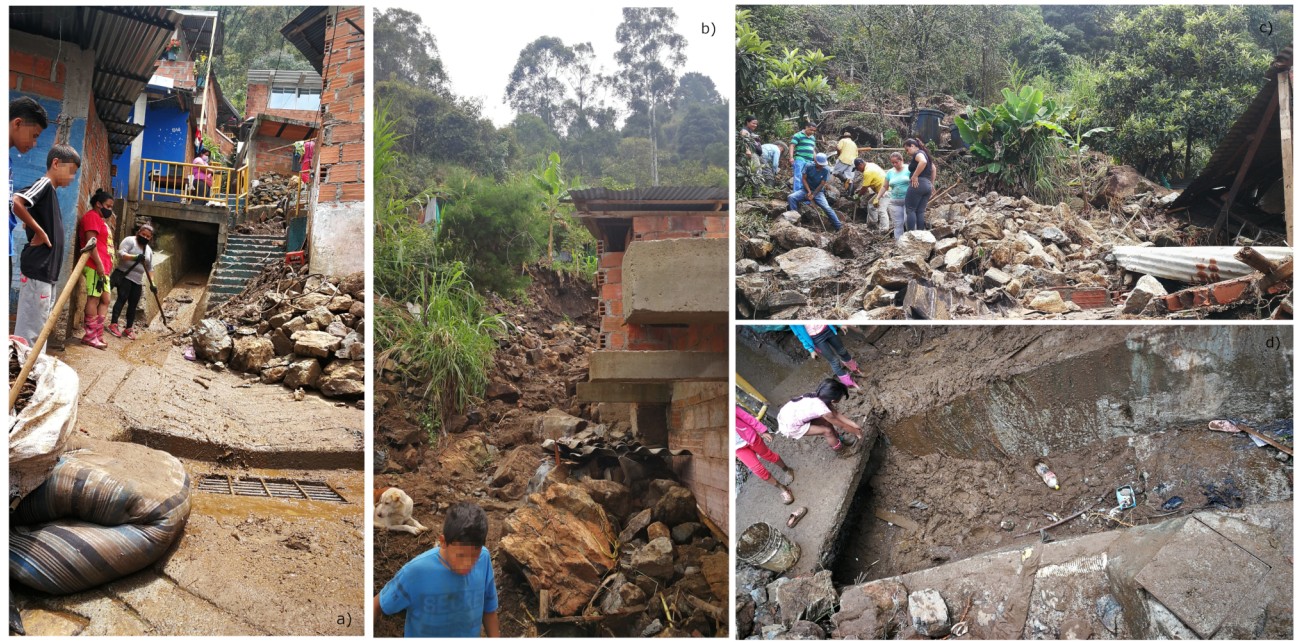

**Figure 3.** Pictures from the 2021 La Rafita Creek Flash flood event in El Pacífico. a) cleaning process of mud and debris by the community. b) household affected by the flood. c) Upper part of the neighborhood after the flash flood. d) mud in the lower portion of the neighborhood. Photos: Junta de Acción Comunal El Pacífico

2019. From 2004 to 2021, DAGRD responded to 35,827 natural disaster events. In the Aburrá Valley Metropolitan Area, there is evidence of critical rainfall, and rainfall accumulation indices over periods of 1 to 15 days triggering landslides (Guerrero and Aristizabal Giraldo, 2019). These conditions are exacerbated by the nature of the buildings in poorer areas, which are often

located in more marginal, high-risk zones. Therefore, understanding the physical vulnerability factors that influence the risk to the population is essential for providing insights and resources for disaster risk management.

Catastrophic events have been part of El Pacífico's collective memory since its early settlement. Between 2000 and 2001, early in the community's development, they were exposed to both, rockfall and a flash flood from La Rafita Creek (Figure 1), with the flood impacting several homes. From 2012 to 2017, there were rockfall events, though they fortunately caused no

harm to people. However, in 2021, a flash flood from La Rafita Creek destroyed seven homes, caused partial damage to three more, and resulted in eviction orders from the municipality for over 52 dwellings. Figure 3 shows photographs taken by the community during the 2021 flash flood event in the neighborhood.

## 2.2   Physical vulnerability assessment

For the purposes of this work, we consider a *'hazard'* to be a natural process such as flooding, landslides, and rockfalls which

may or may not be exacerbated by anthropogenic actions. When assessing *'exposure'* we primarily consider the location of buildings, and when considering *vulnerability* we determine any preexisting characteristics of the built environment in the





community (Cardona, 2011; Lavell et al., 2012) that may make them more or less vulnerable to hazards. By independently assessing these different factors we can assess the physical vulnerability of the community.

Physical vulnerability can be quantitatively and qualitatively assessed in a range of ways, such as: 1) Developing fragility
or damage curves that provide the conditional probability that a given building will reach or exceed a certain level of damage severity as a function of the hazard intensity (Peduto et al., 2017; Menoni, 2006). 2) Damage matrices, which link observed damage to specific levels of hazard intensity (Menoni, 2006). 3) Hazard indicators which reflect the hazard and vulnerability of individual buildings. This is calculated by standardizing any quantitative data, ranking criteria, applying a weighting factor, and then summing all components (Papathoma and Dominey-Howes, 2003; Dominey-Howes et al., 2010; Dall'Osso et al.,
2010; Kappes et al., 2012).

In the case of El Pacífico much of the information regarding the engineering of buildings and the hazard intensity and frequency was unavailable. Instead, the primary data source was the community's knowledge of the hazards they are exposed to and their impacts. With this in mind, we developed a participatory methodology to assess community vulnerability to hazards. Developing community risk assessments is a technique that is commonly employed by NGOs and community-based
organizations, especially within the field of climate change adaptation (van Aalst et al., 2008), where it is recognized that bottom-up approaches can be more impactful than top-down policies. Previous studies have highlighted the need for community members to be embedded in the co-production and co-synthesis of knowledge so that learning from the processes can be beneficial to both researchers and community members (Fazey et al., 2010). Developing qualitative indicators with community members in an iterative process with researchers to ensure that they are maximally accurate, reliable, and sensitive is therefore
highly beneficial (Reed et al., 2006). Employing a participatory approach to integrate interdisciplinary knowledge ensures that the resulting tool is useful according to the needs of the end users and also increases the acceptance of the resulting vulnerability maps (de Brito et al., 2018). This ensures that the end users (the community of El Pacífico) are confident in using the outputs from this study to advocate for themselves and are fully aware of the inherent uncertainties of the data sets before they rely on them for decision-making purposes. We use this approach to create physical vulnerability assessments for a range of hazards
(flood, landslide, rockfall) for the community of El Pacífico.

## 3 Development of participatory vulnerability assessment

The main objective of this study is to establish the physical vulnerability of the community to three types of risks (rock falls, landslides, torrential floods) in an urban self-build settlement in the city of Medellín. This is based on the co-construction and analysis of historical information of events and community perception of risk, to generate useful hazard information with
an adequate spatial scale and a detailed buildings dataset. This approach is described below. Firstly, we establish some key concepts about risk and physical vulnerability as the presented in this study and secondly, we present a novel methodology for assessing hazard and physical vulnerability.



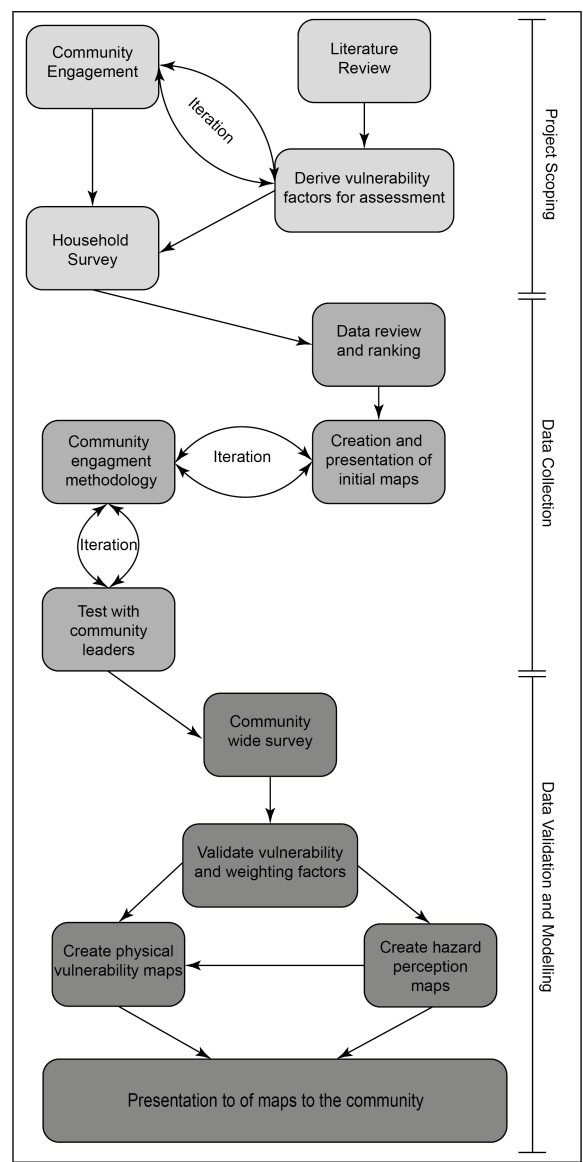

**Figure 4.** Stages of the methodological workflow for the co-construction of physical vulnerability maps for the El Pacífico community.

The methodology for assessing the physical vulnerability of a community based on physical vulnerability indicators and community-generated data is formed of three stages (Figure 4), namely: Project scoping, Data collection, and Data validation and modeling





## 3.1 Project scoping

In creating the methodology for generating the physical vulnerability assessment we focus on three main areas: a review of previous methods, the derivation of appropriate vulnerbaility factors and iterative community engagement. El Pacífico is a close-knit community that has a well-organized structure. They have previously conducted risk assessments and have a 140 particularly engaged with risk studies. Indeed, this project recognises the importance of key figures as community researchers. This was essential for ensuring equitable community engagement in the project whose outputs are a collaborative effort between all stakeholders involved.

### 3.1.1 Community Engagement

El Pacífico has been actively engaged in research since 2016 when they collaborated with researchers from Colegio Mayor de 145 Antioquia (Colmayor) to conduct a community census to establish a baseline for the provision of public services in households within the neighborhood (Agencia IPC, 2016). Later, they participated in a community risk reduction program and helped define risk scenarios to create a community action plan (Grupo de Investigación Ambiente Hábitat y Sostenibilidad et al., 2019). As a consequence of this, the community already identified their needs with respect to knowledge gaps. They were specifically focused on the risk posed to the community by floods, landslides and rockfalls and that this information was 150 required at the household/building scale. Therefore, we determined that data collection would need to focus on a building survey and records of past hazard events, both conducted by the community researchers and iterated with key community leaders.

### 3.1.2 Literature review & Derive vulnerability factors for assessment

Before conducting the building survey it was essential to define relevant vulnerability factors that would form the basis of our 155 analysis. We therefore conducted a literature review (Kappes et al., 2012; Silva and Pereira, 2014; Papathoma-Köhle et al., 2017; Singh et al., 2019; Pereira et al., 2020; Rodríguez-Gaviria et al., 2019) to identify the appropriate variables. These focused on criteria that would change the vulnerability of a building to damage by the three different hazards that are the focus of this study. For example, a building made of concrete will likely have a lower vulnerability to landslide events than a building made of wood. These vulnerability factors were iterated with the community to ensure that they fully captured their areas of 160 interest (Table 1).

## 3.2 Data Collection

As part of the wider URBE Latam project an overarching household survey was conducted (Figure 5). The community in El Pacífico is not represented on the official census of the area so collecting this data is important to ensure representation of their needs. As this survey was already planned we were able to add additional criteria that represented the information required for 165 the physical vulnerability assessment (Table 1).



**Table 1.** Physical Vulnerability Indicators defined with the community and the literature review

| Indicator | Hazard | Potential Values |
|---|---|---|
| Building material | Landslide, Flash flood, Rockfall | Brick<br>Concrete block<br>Wood, Bamboo<br>Mixed materials<br>Other |
| Number of floors | Landslide, Flash flood, Rockfall | 1<br>2<br>3 |
| Building reinforcement | Landslide, Flash flood, Rockfall | Yes/No |
| Roof structure | Rockfall | Clay tile<br>Zinc tile<br>Asbestos tile<br>Concrete slab<br>Wood roof |
| Floor material | Landslide, Flash flood, Rockfall | Marble<br>Porcelain tile<br>Concrete<br>Wood<br>Dirt |
| Distance to the slope | Landslide, Flash flood, Rockfall | Distance (m) |



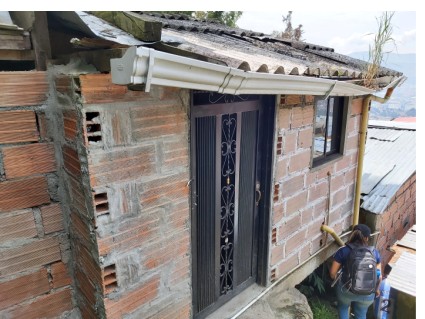 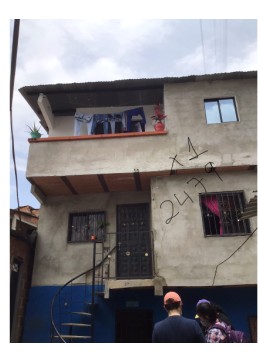 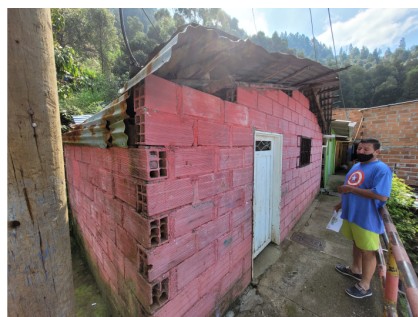

**Figure 5.** Data collection during the household survey left) one-story household. center) three-story household. right) one-story household. Photos: Urbe-Latam survey team.

### 3.2.1 Household survey

The survey was co-created with the community and had five main sections: i) consent, ii) general information of the household (telephone, address, sector), iii) family info (for the use of the community census), iv) household information (5 first rows of Table 1) and, v) information of previous events (the document is included as Annex **??**1). The survey was undertaken with a geographical disaggregation of the neighborhood in five sectors following the proposal of the community researchers within the community (Figure 6). The research team along with the community researchers undertook the neighborhood collection of the data (Table 1) and took pictures of each one of the surveyed buildings. As a household identifier, we used the provisional number given by the utility company, called "interior", which is used as an address for the billing of electricity, water, gas, and/or internet, as the neighborhood does not have official addresses. Due to COVID-19 restrictions, the survey was done after the most strict phase of confinement ended, the consent included permission to take pictures of each household, and all the data was uploaded to an online repository where it was cleaned and classified.

### 3.2.2 Data review and ranking

Once uploaded to the online repository the data collected from the household survey was reviewed to ensure that the data met a minimum standard. This included reviewing whether answers from the survey matched photographs taken, to limit any bias the inhabitants may have regarding their households or limited knowledge of materials as well as the quality and completeness of each one of the entries.

Pictures were analyzed visually and a numerical score for each indicator was awarded for a range of potential values (Table 1). The survey managed to collect data from 120 buildings in total, with each building scored for each hazard for 6 criteria in Table 1. Weighting factors were defined to reflect the vulnerability of a particular criteria to a particular hazard. For example, how the impact of a landslide might be dictated by the construction materials of a building, with wood behaving less well than brick. These weights were defined by reviewing literature from similar studies (Kappes et al., 2012), and then were reviewed by the in-country partners to ensure that they were appropriate, given their knowledge of the community-building practices





**Table 2.** Weights of indicators. In bold there is the relative weight of each indicator within each hazard.

| Indicator | Category | Hazard | | |
|---|---|---|---|---|
| | | Landslide | Torrential Avenue | Rockfall |
| Building material | | **0.25** | **0.15** | **0.1** |
| | Brick | 0.5 | 0.5 | 0.4 |
| | Mixed Materials | 0.8 | 0.6 | 0.6 |
| | Wood | 1 | 1 | 1 |
| Building Condition | | **0.25** | **0.2** | **0.1** |
| | Good | 0.2 | 0.3 | 0.3 |
| | Acceptable | 0.5 | 0.5 | 0.5 |
| | Poor | 0.8 | 0.7 | 0.7 |
| Building reinforcement | | **0.15** | **0.15** | **0.15** |
| | Yes | 0.3 | 0.2 | 0.3 |
| | No | 0.8 | 0.7 | 0.8 |
| Number of floors | | **0.05** | **0.3** | **0.05** |
| | 1 | 1 | 1 | 1 |
| | 2 | 0.5 | 0.5 | 0.5 |
| | 3 | 0.3 | 0.3 | 0.3 |
| Roof Material | | - | - | **0.3** |
| | Asbestos Tile | - | - | 0.8 |
| | Zinc | - | - | 0.8 |
| | Concrete | - | - | 0.2 |
| Roof Condition | | - | - | **0.15** |
| | Good | - | - | 0.2 |
| | Acceptable | - | - | 0.5 |
| | Deficient | - | - | 0.7 |
| Number of rows from the slope | | **0.3** | **0.1** | **0.25** |
| | 1 | 1 | 1 | 1 |
| | 2 | 0.7 | 0.9 | 0.7 |
| | 3+ | 0.5 | 0.6 | 0.3 |
| Number of rows from the channel | | - | **0.1** | - |
| | 1 | - | 1 | - |
| | 2 | - | 0.7 | - |
| | 3+ | - | 0.6 | - |

(Table 2). In this sense, there were several interactions with the community to guarantee data quality. Creating a dynamic data collection strategy where the quality of data collection per building was independently validated.

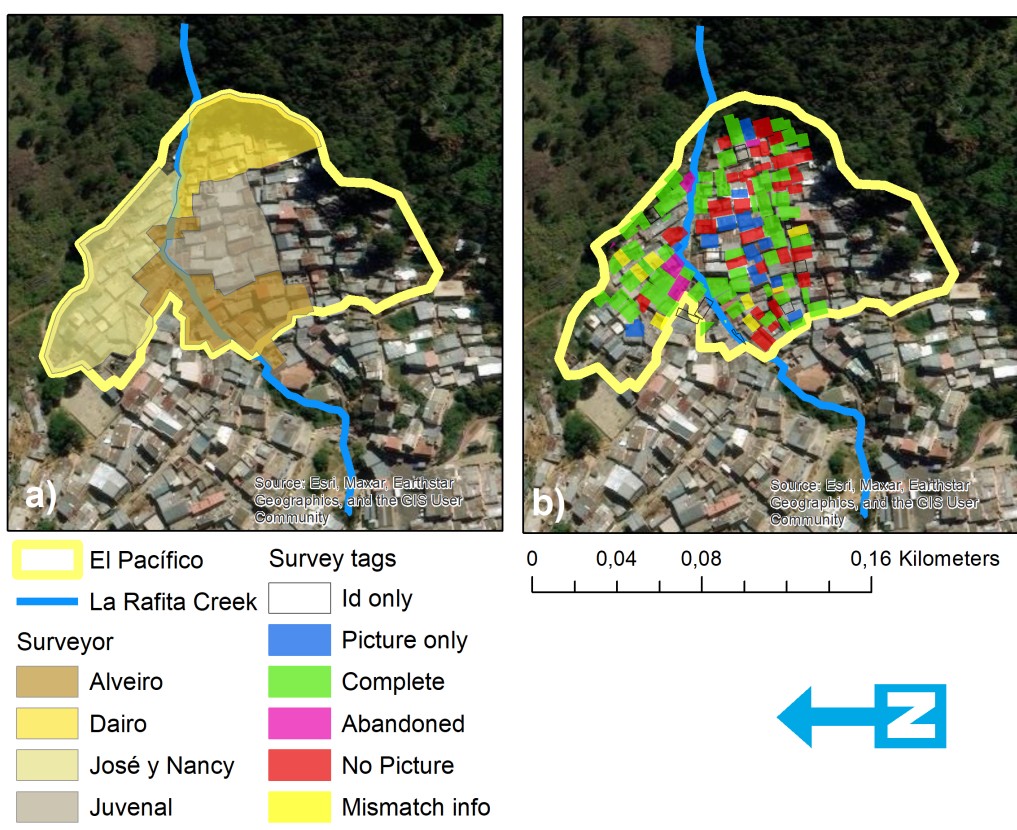

**Figure 6.** a) Survey sector defined by the community leaders. b) Stages of quality control of the household survey dataset

### 3.2.3 Creation of maps

Once the data repository was created a first set of maps was produced. This process consisted of giving a color code to each building in the survey according to its state (Figure 6): i) complete survey with picture (green), complete survey without picture (red), incomplete survey with picture (blue), mismatch information between survey and picture (yellow) and abandoned household (magenta). The community leaders each took responsibility of data collection in specific zones of El Pacífico, the areas of which are described in Figure 6. Complete surveys with households on the map with pictures were reviewed by a team of civil engineers and geologists to confirm the information given by the household dwellers. For those surveys without pictures and mismatched information (yellow and red), data collection was iterated with the community. This phase of data collection integrates the community knowledge and completes the database needed for the physical vulnerability assessment. Figure 5 shows how the research team conducted the onsite data collection.





### 3.2.4 Community engagement methodology

To facilitate cross-disciplinary discussion between different stakeholders and specialists involved in data collection, we developed a prescriptive protocol of what information the community engagement activities should attempt to capture. Within this framework, we endeavored to clearly define the objectives, aims, tools, and data that were required from each component of the community survey. The overarching objective of this community engagement activity and the protocol we developed was to enable dialogue with the community and therefore promote greater understanding of the risk of natural hazards. In consultation with community researchers, we decided that we would produce a booklet for participants to work through. The workbook began with an update of the URBE-Latam project status, an introduction to the workbook and what we hoped to use the data for, and a request for consent to use the data in further studies. The protocol and therefore the booklet were split into 4 distinct sections: Part 1: Hazards, Part 2: Exposure, Part 3: Hazard Interventions, and Part 4: Vulnerability.

### 3.2.5 Test with community researchers

To ensure that the protocol would function was expected we asked the most active community researchers to beta test the first version. Community leaders then gave feedback, which focused mainly on simplifying technical language and the order of questions / sections. The primary concern raised in this session was that there would need to be technical team members involved in the dissemination of the survey in case explanations were needed during the engagement.

### 3.3 Data Validation and Modelling

The community-wide survey was completed with community researchers, leaders and inhabitants using the following protocol:

**Part 1: Hazard** – This section aimed to understand if current hazard assessments for the area align with the community's perception of hazards. The existing hazard assessment that covers El Pacífico was created for the city of Medellin and as such does not include the fine detail that members of this community can provide. For this task, the hazards addressed were pre-selected based on previous interaction with community leaders. In the case of El Pacífico, this included: Flood, Torrential Avenues (a local term for a hyper-concentrated flood event), Landslides, and Rockfalls. Each of these hazards was defined in the workbooks, and example pictures were provided so that there was no confusion with terminology . The community members were also provided with multiple blank base maps of the settlement , and were asked to draw polygons that they felt defined the likely extent of these hazards.

For each hazard, participants were asked to consider whether they felt that the community ever experienced these events and if so they were asked to identify areas of high, medium, and low risk. To facilitate this segregation of the hazard they were asked to define areas where they have ever known of a hazard occurring, and then with a different color, identify areas where the same hazard occurs regularly i.e. yearly. From this description, we assume for this study that areas that are identified as frequently being exposed to a hazard are high-hazard areas, areas that have ever been exposed to a hazard are medium-hazard areas and the areas that were identified as never having suffered from a hazard are low hazard areas. This is, of course, a simplified characterization of the hazard and perhaps underestimates the impact of low frequency, high magnitude events but





to translate the community knowledge we felt that this approach captures some granularity in the characterization of hazard that would otherwise be missing. For larger events that would likely impact the entire community, the existing lower-resolution hazard maps communicate the overall hazard potential.

Finally, participants were asked if they had experience with these types of hazard events and to identify on the map the location of these events and record any damage that was associated. Participants were then asked questions related to the frequency of these hazards relative to each other as well as the likely impact of each hazard relative to each other. Here, we wanted to explore community perception of the frequency magnitude relationships of these hazards, which gives us more information for tuning the weighting factor for each relative vulnerability map. To capture this information participants were

asked to rank hazards relative to each other for frequency and potential damage, they were also asked to explain their reasoning for the final ranking.

**Part 2: Exposure of significant collective infrastructure** – This section of the community engagement aimed to increase our understanding of the key infrastructure that the community may: a) use as a service, b) use as a point of congregation, c) be dependant upon for social/physical/cultural or religious welfare. Whilst the outcome of this exercise will not play a part

in the computation of the final model, it will help to correlate high-vulnerability buildings and areas that intersect with the community's other needs. From here the community can make decisions about whether they wish to re-locate any of these key services.

For this section of the community engagement, the participants were given a fresh, blank paper map of the community and asked: Which buildings are most important to the community as a whole? Why is this? Locate the buildings on your map and

annotate the function (i.e. school, shop, church, etc). Are there other physical assets/spaces within the community that you regard as important? Locate these on the map and describe their function and value. This prompted discussion of how hazards may generate a negative social impact and what services or social interactions would be restricted as a consequence of this disruption.

**Part 3: Ideas for hazard interventions** – In this section we assess the community awareness of processes that increase or

decrease the hazard potential of landslides, floods, and rockfalls. Is there a clear understanding for example, that decreased infiltration will lead to flashier flood responses or that deforestation on slopes above the community is likely to increase the landslide potential? Understanding the community's view of these processes strengthens their capacity to perform interventions to manage the effects of these processes. Their increased awareness will also enable them to fully advocate their key priorities to local policymakers.

This section of the community engagement was recorded as a discussion where facilitators encouraged attendees to discuss whether there were any processes or practices that they felt might make hazards better or worse within their community. They were also encouraged to review the outputs from Part 1 and Part 2 of this exercise and consider any locations that are exposed to multiple hazards and contain key elements at risk or else are important spaces to the community. These would be areas that in the future would benefit from resource prioritization. They were also asked to comment on whether or not they found any

of the results from the previous sections surprising if there were areas that seemed to be higher risk than they had previously considered.


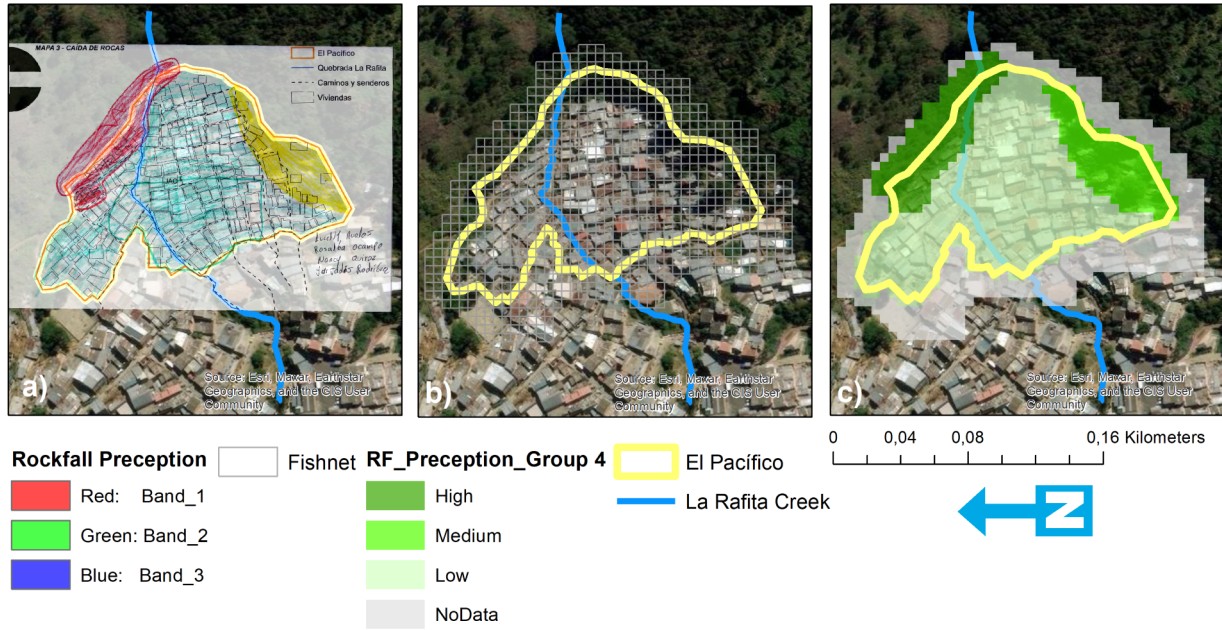

**Figure 7.** a) Map of rockfall hazard drawn by group 3 during the completion of the data collection protocol workshop. b) 10mx10m grid overlapped with the perimeter of the neighborhood. c) Rockfall hazard perception drawn by group 3 in gridded format.

**Part 4: Vulnerability** (validating vulnerability and weighting factors) – As part of the original data collection process we assessed the building vulnerability in regards to the building materials, building condition, building reinforcement, number of floors, and state of the roof (see details above). This assessment, coupled with a survey of the literature allowed us to derive several weighting factors to use in the model to represent the vulnerabilities of different buildings to each hazard. We felt however that, as with the hazard assessment, it was important to understand and quantify the community's perception of these different factors to the building's vulnerability. In this section we therefore asked inhabitants and leaders to give weights regarding the vulnerability of given materials, conservation, and location of buildings already classified by the community and academic researchers. Through this process, we analyzed the potential biases from inhabitants and researchers and validated the robustness of weights in the communities perception.

### 3.3.1 Create Hazard Perception Maps

To create the hazard perception maps, 16 senior members of the community including the community researchers generated hand drawn hazard maps for each hazard (Figure 7a) and asked to give a qualitative classification of each hazard as high, medium, or low. They then worked in small groups to construct a new consensus map that gathered the perception of each member of the group.


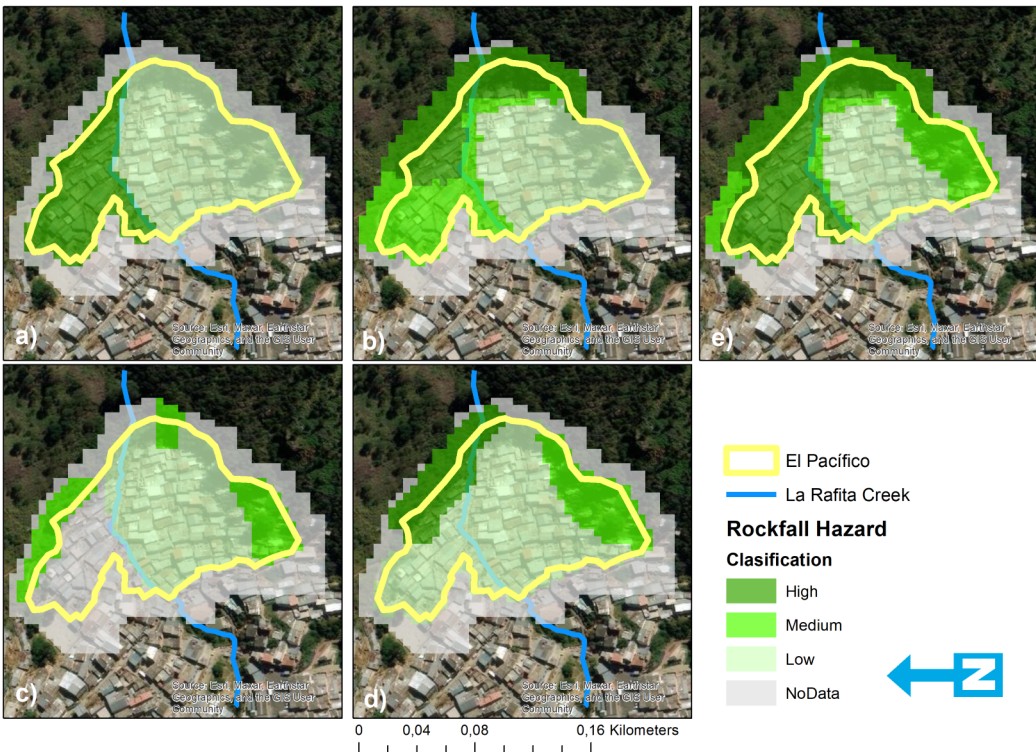

**Figure 8.** a) Rockfall hazard perception for Group 1 b) Rockfall hazard perception for Group 2 c) Rockfall hazard perception for Group 3 d) Rockfall hazard perception for Group4 e) Worst-case scenario for the Rockfall hazard perception

We therefore obtained four maps that were the result of the consensus between four senior leaders each (Fig 8 a, b, c and d). The hand drawn maps were digitised by applying a common grid of 10x10m. To produce a final hazard perception map we defined two approaches: i) the median map, which is the median between all the qualifications, and ii) the worst-case scenario map, which is produced with the least favorable qualifications in each one of the grids. Figure 8e presents the result for the
worst-case scenario for the rockfall hazard perception. Is worth noting that the selection of the worst-case scenario was the result of an effort to guarantee the community safety, nevertheless, we analyzed also the residuals computed as the worst-case minus the median-case (Figure 11) as a sensitivity approach, acknowledging the drawbacks related to index-oriented methods not only in regards to data dependency but also due to representation, normalization and presenting of results (Papathoma-Köhle et al., 2019).
Figure 9 shows the three hazard perception maps formed by community consensus. With this information, the community obtained a new high-resolution dataset of hazards that helped them and the local authorities to better plan further mitigation or interventions in the neighborhood. As for each one of the studied hazards, the El Pacifico community found the next: i) the central portion of the neighborhood is under a high torrential avenue hazard, mainly in the La Rafita creek channel (Fig. 9 left); ii) regarding rockfall the northern and northeastern portion of the neighborhood poses the highest hazard (Fig. 9 center), and




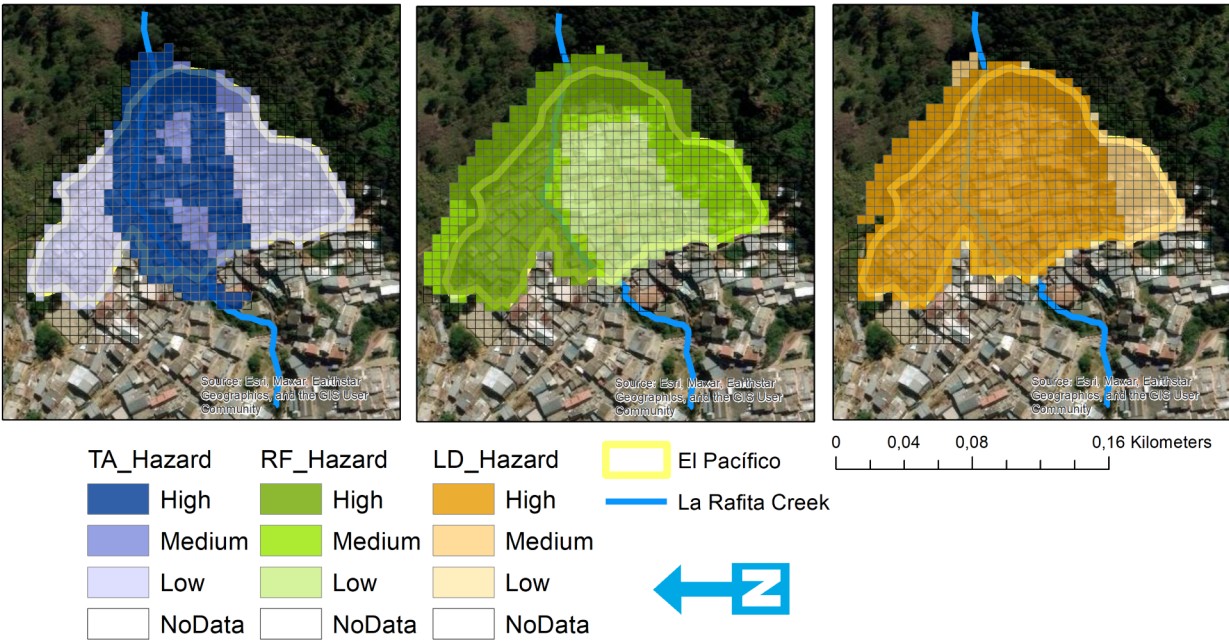

**Figure 9.** Torrential avenue (left), rockfall (center) and landslide (right) hazard perception maps for the worst case scenario

iii) for a landslide the community has a sense of hazard as nearly 80% of the neighborhood is classified with high hazard (Fig. 9 right). It is important to note that the results presented here regarding hazards are the worst-case scenario ones, as a result of the consultation between the community and the research team.

### 3.3.2   Creation of Physical vulnerability Maps

Physical vulnerability maps were generated by combining the hazard perception outputs with the data collected from the
household survey concerning exposure and vulnerability, integrated spatially. The following formula was used to aggregate all data. This was done using Excel spreadsheets as the a) the data sets were small and b) using a straightforward, easily available program makes it easy to train the community to input data as they produce it and therefore enables them to update these maps as and when they have added to the data production.

    Each hazard is treated separately, creating a relative vulnerability map where hazards are comparable.

$$RHpv_i = L_i \left( \sum HBm_w HBc_w HBr_w L/TaBf_w RfRc_w HRo_w TaRc_w \right) \qquad (1)$$

    Where:

    $RHpv_i$ is the relative physical vulnerability of location $i$ relative to the hazard





$H_i$  is the hazard derived from community perception at location $i$

$HBm_w$  is the weighting factor related to how building materials are expected to withstand the specific hazard

$HBc_w$  is the weighting factor related to how the condition of a building is linked to the potential damage the hazard could do

$HBr_w$  is the weighting factor related to how the reinforcement of a building is related to how that building will respond to the hazard

$L/TaBf_w$  is the weighting factor linking the number of floors in a building to how that building is likely to respond to a landslide or a torrential avenue

$RfRc_w$  is the condition of the building roof (only relevant for rockfall).

$HRo_w$  is the number of rows the building is from the hillside behind the community.

$TaRc_w$  is the number of rows the building is from the river channel that flows through the community (only relevant for torrential avenues).

In Figure 10 we present the results of the estimated physical vulnerability, understood as the result of the interaction be-
tween the household characteristics and the underlying hazard following the vulnerability indicators approach described in the methodological framework. For each hazard, we computed the physical vulnerability in each household and we can see that: i) for the torrential avenue both the southern and northern sides of the La Rafita Creek are areas with higher physical vulnerability, being higher in all the southern portion (Fig. 10 a), ii) for the rockfall the higher physical vulnerability is found in the eastern part of the neighborhood, next to the mountain slope, being higher in the northeastern slope and north side of the La Rafita
creek (Fig. 10 b), and iii) for a landslide the results show that in general, the household conditions of the neighborhood induce a mid to high physical vulnerability (Fig. 10 c). Hazard perception maps along with Physical vulnerability maps are used as an aid for decision-making in the community, for that purpose the URBE Latam project developed instruments (Figure 13).

Regarding the decision to consider the worst-case scenario for calculating physical vulnerability, Figure 11 illustrates the residuals computed as the difference between physical vulnerability computed with the worst-case hazard scenario and that
from the median hazard scenario. This outcome highlights the significant variability in hazard perception within the southern section of the neighborhood. This variability is particularly pronounced for landslides (Fig. 11). The uncertainty for landslides can be attributed to their infrequent occurrence in the neighborhood. Consequently, there is a lack of consensus among the residents who participated in the hazard perception exercise.

### 3.3.3 Presentation of maps to the community

The final step in the co-creation of the physical vulnerability maps was that they were presented to the community for their review. Community leaders led presentations to the wider community, regarding the process of developing the maps. These sessions were interactive and provided the community further opportunities for feedback. The reception of these maps was

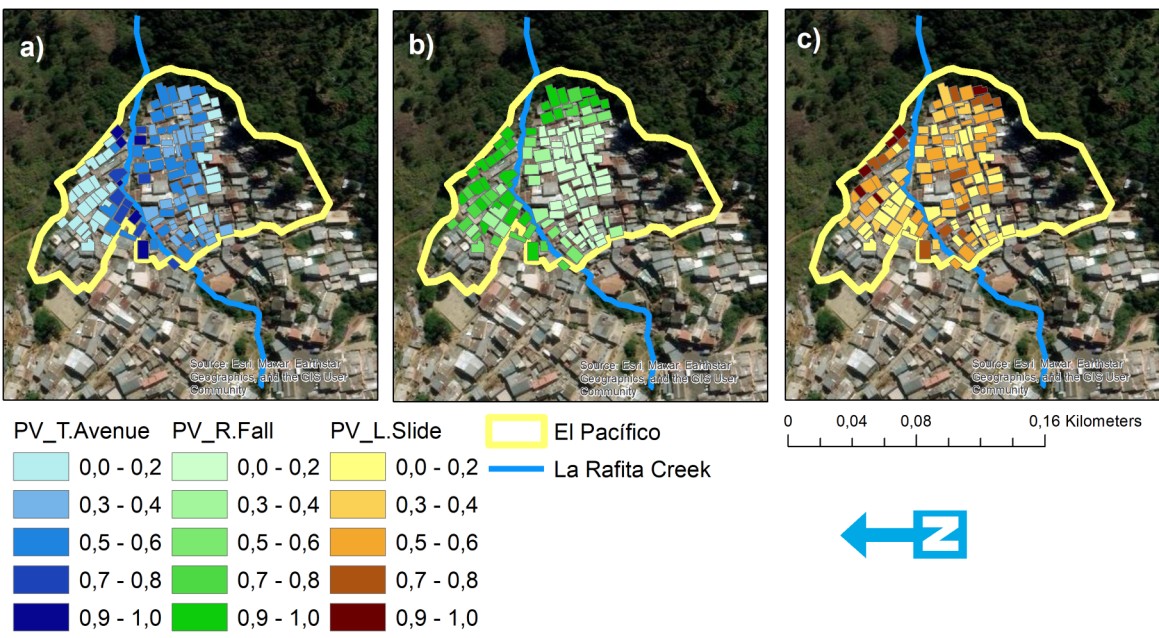

**Figure 10.** Torrential avenue (a), rockfall (b) and landslide (c) physical vulnerability maps for the worst case scenario

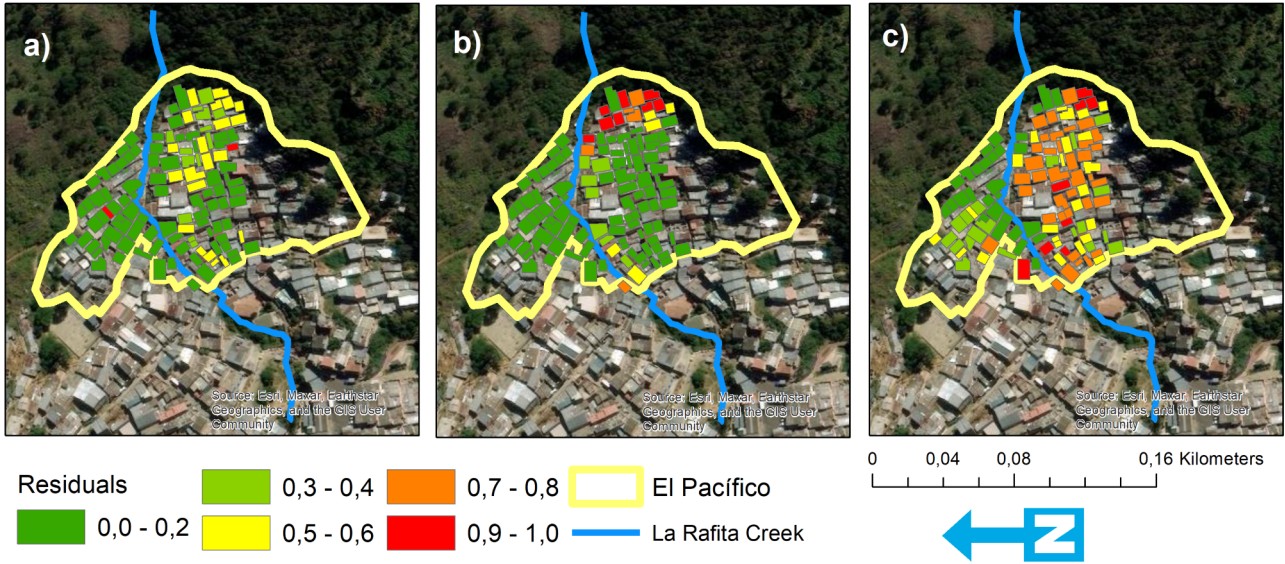

**Figure 11.** Residuals computed from the difference between physical vulnerability at the worst case scenario minus the median scenario for Torrential avenue (a), rockfall (b) and landslide (c)





positive, with the communinity elecgting to adopt and hang the maps on the walls of their community center (Figure 9). Their engagement as key to the co-creation of the data created a strong sense of ownership over the final product, as well as the ability

to continue to add to them. At its core this study aimed to co-produce knowledge that would increase community awareness of hazard and vulnerability, to empower them to make decisions that could lead to reducing their vulnerability and to give them the tools to advocate on their own behalf during negotiations with local authorities. With this in mind, the community selected the outputs that most suited their needs.

## 4 Discussion

The community of El Pacífico approached this study with their own aims, specifically to reduce the communities vulnerability to natural hazard events and to be empowered to advocate for themselves to local decision makers. With this in mind, we have created a participatory mapping approach that allows the community to be at the heart of the data collection and interpretations methodology. This tunes the outputs to their specific needs and has ensured that they have been able to continue to grow their understanding of their physical vulnerability to hazards. All of the outputs presented here have been put to practical

use by the community, and are publicly displayed in the community center (Figure 12). Their intention was to: 1) showcase their progress in mapping physical vulnerability and hazard in their community, which has already garnered the interest of neighboring communities who wish to replicate this within their territories, 2) to better represent the relationship that they have with specific risk scenarios that official maps simply do not showcase due to their scale. Furthermore, the maps serve as a positive reminder of their resilience, history, and collective relationship with disaster risk management which has led to a

self-legitimization of the neighborhood. The process also called the attention of national and international media not only as a community disaster risk management practice but also as a climate change planning strategy (Monsalve, 2023; Telemundo, 2023).

We observed that when asked to define the hazard in their area, the community researchers sense of "memory" was key to their perception, in the case of a flash flood (Figure 11a) there is a higher agreement between the worst-case scenario and the

median-case, revealing that recent past events (i.e. the flooding event of 2021) generated the clearest sense of the possible high hazards. Comparatively, for rockfall and landslides, there is higher variability in the residuals (Figure 11 b and c) as these kinds of events are rare in the community and some of the participants in the construction of the hazard perception maps haven't ever experienced one.

Below we present the positive outcomes from the application of our methodological approach, this evidence was drawn from

informal conversations with the community leaders and researchers and, is presented in quotes:

– **Disseminating and upscalling risk reduction practices**: The community of El Pacífico, as a consequence of their increased understanding, received requests to run a 'Risk school' from surrounding communities. The publication and promotion of their maps increased interest in these tools locally and consequently the community leaders in El Pacífico have received requests to teach others how to enact this process. As they were key to the creation of this knowledge the

are extremely well placed to led knowledge exchange in their local area. The further use of these tools and updating of


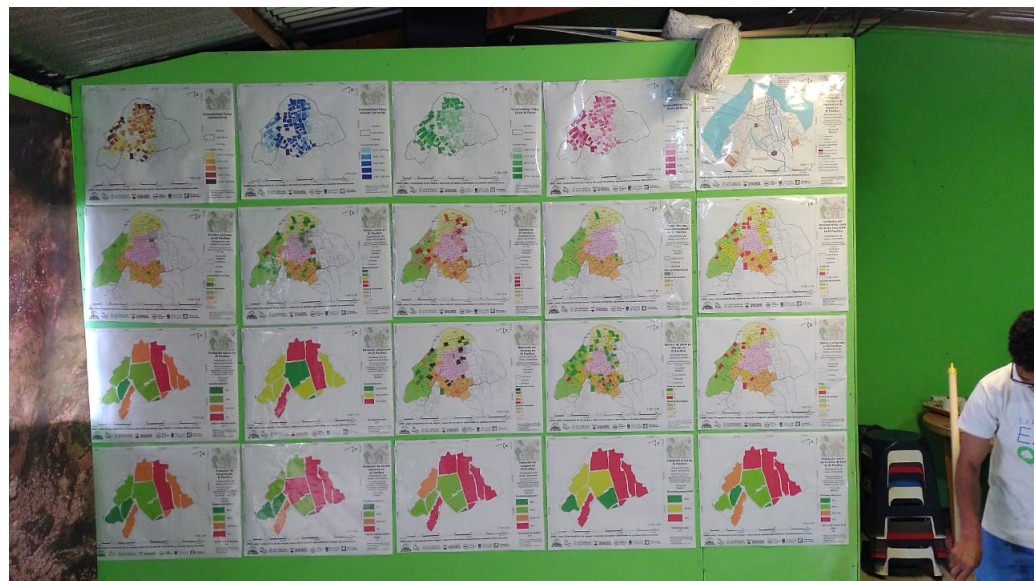

**Figure 12.** Maps as wallpaper at the community center. Photos: Junta de Acción Comunal El Pacífico

the outputs will ensure that this skill set will not be lost over time. The following quote provides a community perspective on how they see the 'Risk school' progressing.

"The neighborhood is considered a trailblazer in the city. They are not only leaders within their community but also represent and guide new initiatives in the commune. They hope these initiatives will expand not only throughout the city but across the country. The Colombian Agency for Disaster Risk Management (https://portal.gestiondelriesgo.gov.co/) has shown interest in their experiences. We are striving to involve more members of the community in the new version of the risk management school. The interventions to reduce vulnerability have been mobilizing more community members, and the attention gained not only within the city but also nationally and internationally has been beneficial not only for the community but also for others. We hope that our methodology becomes an exemplary model for other communities to follow."

– **Usage of Maps**: At the onset of this study, one of the key concerns in El Pacífico was that the existing available hazard maps were not of sufficient resolution to support community decision making. In creating these maps the community wished to highlight to local decision makers a high level of understanding of their needs and the potential interventions that would lower their vulnerability to hazards.

"Maps serve as useful references and can be shared with other communities to encourage them to adopt similar exercises. They are also beneficial in demonstrating to the municipality that the neighborhood is prepared for potential challenges. However, the most crucial aspect is that the maps represent the methodologies and knowledge acquired through hard work."



- **Monitoring change in vulnerability and risk**: The researchers in this study were careful to highlight that the maps that have been produced represented relative vulnerability. The community understand that as they add more buildings to the study and make interventions to buildings in the study, the maps that they have produced will change. This was initially of some concern to the researchers as we anticipated this might undermine the communities faith in the outputs of the study. We found however, the community to be incredibly receptive to the concept that the profile of their vulnerability will change through time, indeed it is their intention to enact interventions to ensure that this is the case. Figure 13 shows the poster that has been produced to explain physical vulnerability to the community.

  "Yes, we are aware that the maps will change with each intervention in the neighborhood, particularly those related to physical vulnerability. For instance, Figure 12 includes building DZ2, which is already undergoing improvements with better materials and reinforcement. Additionally, some houses with high physical vulnerability, located at the edge of the slope, are being demolished and replaced with orchards. As we implement actions to improve food autonomy post-pandemic, we believe that the knowledge gained will help us maintain the results over time. We hope to achieve this either independently or through collaboration with universities and researchers with whom we have established strong ties of cooperation over the years."

- **Supporting community-led risk reduction**: In this study the hazards were selected by the community, those which caused them the greatest amount of concern. The is a high level of understanding in El Pacífico however, that the impacts of hazards such as these are likely to increase as the effects of climate change become more marked. With this in mind the community are already planning to add further hazards to their assessment.

  "Currently, we are focusing on managing the hazards that concern us the most and have been identified on the maps. We are taking measures to prevent rockfalls using local scale slope reinforcement, constructing speed reducers for the creek, demolishing buildings with the highest physical vulnerability, growing orchards in areas of high hazard to prevent new construction, and the installation of emergency signaling. However, it is essential to address new hazards, such as those arising from drought due to El Niño and climate change, as well as potential risks associated with new and taller buildings that may have heavy reinforcement but lack sufficient soil resistance or structural studies. To prepare and support the community's well-being, we must consider these emerging hazards."

The level of trust between the community and the research team was a key factor for the overall success of this project. Without the relationship that the community had built since 2016 with researchers (both national and international), it is doubtful that they would have been so open to co-constructing the methodology presented. Likewise, academic actors were able to have reliable trust in the community given their past experiences with disaster risk management research. The communities ability to adapt to the inherent uncertainties of the hazard and vulnerability data increased the comfort of researchers in communicating and exploring these complex issues. In this environment of mutual respect the outcomes of the project have had impact far beyond the end of the initial study. This suggests that long-lasting relationships based on trust can be a determining factor for the success of similar research in the future, especially in the face of disasters such as the ones seen in the territory as well as unexpected global events such as pandemics.





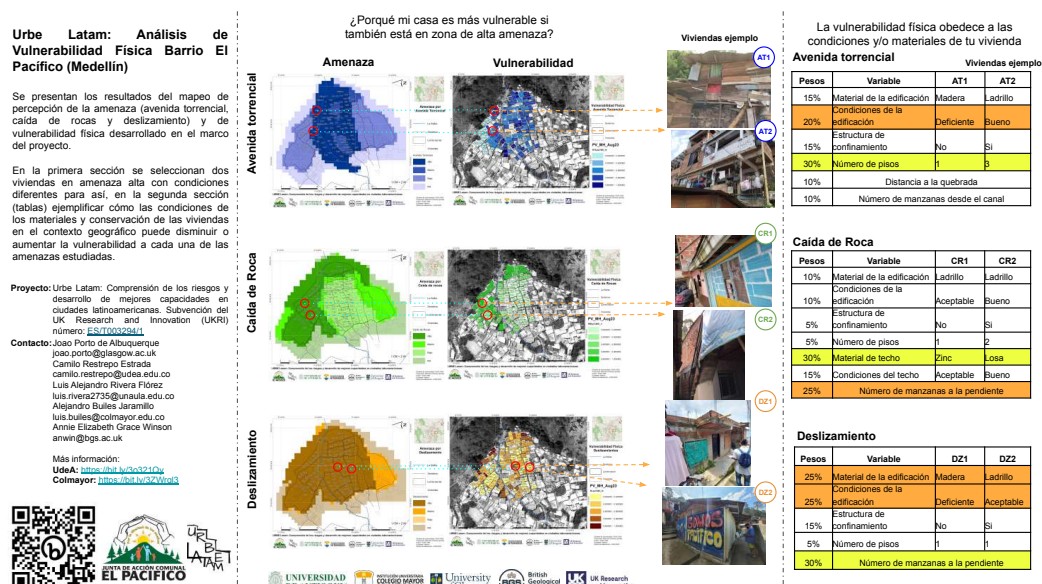

**Figure 13.** Physical Vulnerability explanation poster as a tool for household intervention planning in the community

These outcomes clearly show that our framework enabled the engagement of community members in data generation which not only succeeded in filling evidence gaps in existing official data but also triggered further positive effects by enhancing the critical consciousness about problems and supporting the exploration of solutions to reduce and manage disaster risks. This echoes recent studies (Porto de Albuquerque et al., 2023; Fraisl et al., 2022; de Sherbinin et al., 2021; Rodríguez-Gaviria et al., 2024) and confirms the benefits of participatory approaches to disaster risk reduction centered on community-generated data. Our study adds to these previous studies with a specific and validated methodological framework for vulnerability assessment.

The legal framework to strengthen and improve the usability of citizen-generated data exists, but something is still missing. Even when there is a policy change that should benefit these efforts, such as the climate change emergency act in Medellín (Municipio de Medellín, 2020), or the public policy for residents' protection (Municipio de Medellín, 2021), these do not seem to translate into tangible results for self-built communities (Ulbrich et al., 2023). A good contribution perhaps could be aimed at thinking beyond the classic notions of "lack of political will" or "lack of resources", and perhaps think more in terms of the unwillingness of governments to not only think but also act upon the possibility of disasters. Disaster Risk Management does not make for appealing political proposals to the general public, since it predisposes the political agenda to pursue what-ifs instead of what is traditionally perceived as tangible problems, such as security, economy or education, which make for more appealing political pursuits.

Our methodological framework establishes the basis to replicate this methodology in other communities worldwide (4) by following a participatory approach involving local stakeholders in all phases of the process. First, a scoping phase must be carried out through meetings with the community, identifying their concerns and establishing clear objectives regarding the risks and vulnerabilities to be assessed. Next, a participatory data collection is conducted, where residents as community re-




searchers and leaders collaborate in gathering information about housing characteristics and risk perceptions. This information is supplemented with technical verification through visual inspections and data review by experts. Once the data is collected and verified, community validation occurs, where preliminary results are presented to the residents for feedback and adjust-
ment. Finally, a high-resolution physical vulnerability map is generated, which is used by both the community and authorities to improve risk management, enabling this process to be replicated and adapted in other areas exposed to natural hazards.

Understanding and reducing risk to hazards necessitates an assessment that is attentive to the many ways that communities perceive and respond, cope, and adapt to threats (van Aalst et al., 2008). The importance of a local perspective becomes especially critical in the context of multihazard landscapes, where populations face threats that overlap in time and space and
may interact in complex ways (Gill and Malamud, 2016; Kappes et al., 2012; Mercer et al., 2010) Participatory mapping can provide a means to incorporate local knowledge into vulnerability assessments, potentially revealing vulnerabilities to multiple hazards that are co-produced at local scales.

Participatory approaches include stakeholders in the knowledge production process to incorporate local knowledge in order to provide information that is not readily available from other sources, and is grounded in the experiences of those popula-
tions experiencing exposure to hazards. Many agencies engaged in DRR and CCA have begun to focus on community-based approaches to assessing and reducing vulnerability.

### 4.0.1   Limitations

The process of determining physical vulnerability in the community of El Pacífico was strongly influenced by health and safety concerns due to the COVID-19 pandemic. As a consequence there are some limitations to this study that must be acknowledged
and could be improved upon through time.

Firstly, whilst every effort was taken to create hazard maps with the community, these hazard perception maps have not been validated against recorded hazard events. To our knowledge these inventories do not exist at a resolution that would be appropriate for the community level outputs. However, if independent data sources could be validated against the communities perception maps then it would likely improve the uptake by local response organisations.
The hazard perception maps where also produced by a relatively small number of participants (16). This creates the risk of them being unrepresentative of the communities views. We aimed to mitigate these effects by selecting senior members of the community who we felt would have a good overview of the way that hazards have affected the area. It isn't possible however, to know whether these participants were biased by either their experiences or strong voices within the group. Introducing bias due to small groups is recognized as an inherent risk in the participatory modeling process (Garmendia and Stagl, 2010) as
they often involve only a few participants. This could be further controlled by introducing a formal expert elicitation process. In this case we felt that that this would introduce more decision fatigue into the process and cause the community members to disengage and so we did not include it.

This also means, however that our method as it stands is unable to give a quantification of the uncertainty of the data we have produced. Where possible we have made efforts to perform sensitivity and validation assessments. For example we
compared the median hazard map values with the worst case maps and sense checked our vulnerability assessment through the





community survey. Quantifying uncertainty in an inherently qualitative process is complex, we have attempted to mitigate the effects of this by keeping an honest and open dialogue with the community. They are aware that what they have produced with one possible interpretation of their vulnerability and that this will likely change through time as a more detailed, quantitative understanding emerges.

Finally, we were not able to conduct a full sensitivity assessment of the vulnerability factor weights. This could mean that the prioritization of factors relative to each other is not well defined. We managed this by deriving the weighting factors from other similar studies and by discussing and iterating them with the community. Furthermore we created the final vulnerability maps by mutual consent, allowing all participants to review weighting scheme and come to a consensus on the final hazard perception maps. We have therefore controlled these limitations to the best of our abilities, any further assessment would

require the integration of more quantitative data, which we do not currently have access to.

## 5    Conclusions

The implementation of a participatory methodology for the assessment of physical vulnerability in self-built communities can be a powerful tool, not only for generating vulnerability data at the neighborhood physical scale but also for strengthening community resilience. By actively involving residents in data collection, analysis, and validation, this approach fosters not

only a better understanding of risk but also a sense of ownership of the disaster management process within the community. This methodology, replicable in other urban risk contexts, helps overcome the limitations of official planning systems that, in many cases, fail to capture the complexity of marginal areas. The results obtained not only contribute to improving decision-making and negotiations with authorities but also enhance the community's capacity to manage their territory more safely and equitably in the face of future disasters.

Granularity in disaster risk management must be an important part of territorial management. Spatial abstraction has translated into a broad generalization of risk scenarios, especially in self-built territories. Such generalizations can blur the practices that must be improved upon to strengthen resilience and aid communities in better understanding their risk conditions. For this to happen, trust must become an integral part of risk governance, both from inside and outside communities and local instances of government.

Currently, in Colombia, the laws that define the spatial scale of the most detailed studies for urban planning is defined to be 1:2.000. Such coarse spatial scale is insufficient to address dwelling or even neighborhood scale analysis like the one we just presented. In this sense, our methodology of co-producing hazard and physical vulnerability assessments can become a tool to integrate citizen-generated data and gain more insights by refining the scale of urban planning in Colombia.

The methodology proposed in this study is a result in itself and gives a framework to bridge the gap posed by the quality

assessment of citizen-generated data sets. The feedback between the community researchers and the academia was key for the effective integration of citizen-generated data into the hazard and physical vulnerability assessments.


Junta de Acción Comunal El Pacífico
Personería Jurídica Número 97 de 2009
NIT: 900301828-8

Yo, _________________________, identificado/a con C.C. _________________, en calidad de habitante de El Pacífico, he sido informado/a por la JUNTA DE ACCIÓN COMUNAL (JAC) del barrio sobre el presente censo, el cual tiene como finalidad recolectar información sobre las familias y viviendas del barrio para fortalecer estrategias de evacuación y de gestión de riesgos en el futuro. Acepto que la información aquí consignada puede ser utilizada por la JAC, así mismo que por terceros siempre y cuando estos cuenten con la autorización escrita de la JAC de El Pacífico y se respete la confidencialidad de mi persona y mi familia.

Firma: _________________________________

Fecha: _________________________________

**1. Información general:**
1.1. Encuestador: _________________________
1.2. Sector en el que se encuentra la edificación: _________________________________
1.3. Dirección de vivienda: _________________________
1.4. Teléfono de contacto: _________________________
1.5. Número de vivienda: _________

**2. Información sobre la familia:**
2.1. Número de personas que habitan en la vivienda:

   a) Menores de 18 años de edad: _______     c) Mayores de 55 años de edad: _______

   b) Personas entre 18 y 55 años de edad: _______

2.2. ¿Habitan en la vivienda personas con alguna discapacidad motriz?: _____ ¿Cuántas?: _______
2.3. Número de mascotas en la vivienda: _______

**3. Información sobre la vivienda:**
3.1. Material predominante de vivienda:

   a) Ladrillo ____     d) Mezcla de materiales (Más de uno de los anteriores) ____
   b) Bloque de concreto ____     e) Otros materiales (Plástico, cartón, etc.) ____
   c) Madera ____

3.2. ¿La edificación cuenta con estructuras de confinamiento como vigas y/o columnas de concreto o acero?

   a) Si ____     b) No ____

Junta de Acción Comunal El Pacífico
Personería Jurídica Número 97 de 2009
NIT: 900301828-8

3.3. ¿Cuál considera que es el estado de conservación de su edificación?

   a) Bueno (No hay humedades / grietas / hundimiento) ____     b) Aceptable (Hay al menos una humedad / grieta / hundimiento) ____
   c) Deficiente ____

3.4. ¿Cuál es el material predominante del techo de la edificación?

   a) Teja de barro ____     d) Techo de madera ____
   b) Teja de zinc ____     e) Losa de concreto ____
   c) Teja de asbesto cemento (Eternit) ____

3.5. ¿Cuál considera es el estado de conservación del techo de la edificación?

   a) Bueno (la madera de soporte no tiene deterioro / hay amarre de las tejas-losa de concreto) ____     b) Aceptable (La madera de soporte se está deteriorando / no hay amarre de las tejas) ____
   c) Deficiente ____

3.6. ¿Cuántos años tiene la edificación?: _____________
3.7. ¿Cuándo fue renovada por última vez la edificación?: _____________
3.8. Número de pisos de la vivienda: _____________
3.9. ¿En qué piso habita usted?: _____________
3.10. Material predominante del piso en el que habita:

   a) Mármol, porcelanato, parqué, madera pulida y lacada ____     c) Cemento o mortero ____
   b) Baldosa, cerámica, vinilo, tableta o ladrillo ____     d) Madera burda, madera en mal estado, tabla o tablón ____
        e) Tierra o arena ____

**4. Información sobre desastres:**
4.1. ¿A la edificación han llegado derrumbes, caída de rocas o inundaciones?

   a) Si ____     b) No ____

4.2. En caso de responder si, ¿qué tipo de evento y cuándo?:
4.2.1. Tipo de evento: _________________
4.2.2. Mes (estimado) y año del evento: _________________
4.3. ¿Hubo daños derivados del evento?:

   a) Si ____     b) No ____

4.4. En caso de que la respuesta anterior sea si, en su criterio los daños fueron de impacto:

   a) Alto (La vivienda tuvo afectaciones estructurales en sus muros, cimientos o techo). ____     b) Medio (La vivienda tuvo daños leves en muros, cimientos o techo). ____
        c) Bajo (La vivienda no tuvo daños significativos). ____

4.5. Los daños fueron principalmente reparados de manera:

   a) Autónoma (Asumidos por la familia) ____     b) Colectiva (En convite con colaboración de la comunidad) ____

**Figure A1.** Household survey template used for the project scoping phase

**Appendix A: Appendix A. Household Survey**

**A1**

*Author contributions.* **ABJ**: Conceptualization, Methodology, Data collection, Data Curation, Writing - Original Draft, Writing - Review & Editing. **AEGW**: Conceptualization, Methodology, Data collection, Data Curation, Writing - Original Draft, Writing - Review & Editing. **NQ - DU - JR**: Methodology, Data collection, Data Curation. **LARF**: Methodology, Data collection, Data Curation, Writing - Review & Editing. **EB - CD**: Methodology, Data collection, Data Curation. **CRE - INGM**: Data collection, Data Curation, Writing - Review & Editing. **JPA**: Methodology, Writing - Review & Editing, Supervision, Funding Acquisition.

*Competing interests.* None of the authors have any competing interests.




*Acknowledgements.*   This article is part of the UKRI Global Challenges Research Fund Project URBE Latam: Understanding Risks and Building Enhanced Capabilities in Latin American cities (2019–2022) (GCRF grant: ES/ T003294/1, PI João Porto de Albuquerque). The authors are grateful to the entire project team from the University of Glasgow, the University of Warwick, Institución Universitaria Colegio Mayor de Antioquia, the Universidad de Antioquia, the Universidade Federal de Rio de Janeiro, CEMADEN Brazil, the British Geological Survey and the Banco Comunitário do Preventório. We are especially indebted to our partner communities El Pacífico in Medellín (Colombia)

and Preventório in Niterói (Brazil), who actively co-produced the research results. It is important to note that this project was conducted during the COVID-19 pandemic. So the community was required to be exceptionally adaptable in terms of how they collected data and interacted with partners from other countries, who were not able to travel to them. Without their dedication to complete this project for the benefit of their community this work would have been impossible. Open access policy: For open access, the authors have applied a Creative Commons attribution (CC BY) license to any author-accepted manuscript version arising from this submission.



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
