# Peer review of "Community-driven natural hazard and physical vulnerability assessment in a disaster-prone urban neighborhood"

_Natural Hazards and Earth System Sciences, 2024_

## Author Response (AR1)

**DISCUSSION About manuscript nhess-2024-221**.**

Dear editor and referees.

We thank you for your useful comments and suggestions that are of great importance to improve our work. Next, we present the point-by-point responses to reviewers.

The recommendations are addressed below. Comments in **black** and responses in **blue**.

**RC1**

Thank you for the comment. The paper presents an important gap linking academia, practitioners, and local people. However, numerous physical vulnerabilities via observation surveys are already present. However, verification and validation from the community are often lacking. This an important study which would be valuable. Here are my comments.

1. Please add research questions somewhere before the objectives.

Reply 1: We agree with the useful suggestion and have included in our revised version the objective of the paper in the introduction as well as the knowledge gap we hope to address (lines 50-44), as follows:

To address this knowledge gap, we propose a method for enhancing the indicators approach to incorporate community goals, community information, and a feedback process where community researchers play a key role in the physical vulnerability analysis. We introduce a co-created approach to assess the physical vulnerability of the community to three distinct hazards: landslides, floods, and rockfalls. This approach bridges the gap between current official hazard assessments, which often lack specificity and context, and the community needs through co-designing the methodology for data creation with the community

2. "In this paper, we provide a brief context about the study location (Section 2), outline the methodology that we co-created with the community to integrate perceptions of risk into hazard and vulnerability assessments alongside the final physical vulnerability maps created for each hazard (Section 3), as well as present a discussion of contributions (Section 4), followed by a short conclusion (Section 5)." This may be deleted.

Reply 2: Thanks for the recommendation, we agree that this is something expendable, we will delete it.

3. The paper fails to acknowledge similar studies that have used similar methods to measure physical vulnerability. Notably, FEMA154 (see 10.1007/s42452-019-1681-z) or similar methods (10.1016/j.ijdrr.2025.105206) or more advanced methods. The paper should at least mention some papers.

Reply 3: Thanks for your recommendation. We have included a summary on results from a wider pool of literature in the introduction that includes physical vulnerability, we have included the references you recommend as they perfectly fit in our Literature Review. In section 2.3 we have added the following references:

"Previous studies have highlighted the need for community members to be embedded in the co-production and co-synthesis of knowledge so that learning from the processes can be beneficial to 110 both researchers and community members (Fazey et al., 2010; Xu et al., 2025; Ahmed, 2025)."

4. I see some methods in L105. Can be moved to the Intro/LR section.

**Reply 4: This is a useful suggestion that we implemented.**

5. Figure 4. Hazard/risk perception literature is almost missing. There are studies that have linked physical vulnerability with perception (10.1016/j.ijdrr.2019.101317). Or how hazard maps are correlated with perceived extent (10.1007/s40808-022-01442-2)

Reply 5: Thanks for your comment. We clarified this point in the section on hazard perception, and included new references. We also included a reference to the framework we co-designed with the community for a dialogic data collection and mapping (https://doi.org/10.1080/23311886.2024.2307181), which covers risk perception mapping. Section 2.6.1 on hazard perception maps now includes a selection of references:

"If no hazard maps are available, there are different ways to create maps based on community knowledge (Kienberger, 2014; RUIN et al., 2007; Chowdhooree et al., 2019; Khan et al., 2019; Rasool et al., 2022). In the case of El Pacífico they had been aware of participatory mapping with previous results (Grupo de Investigación Ambiente Hábitat y Sostenibilidad et al., 2019), 275 and for this particular task, we used a dialogic approach based on Milton Santos and developed in the URBE-Latam framework (Rivera-Flórez et al., 2024)."

6. 3.1.2 Literature review & Derive vulnerability factors for assessment. This belonged to the LR section.

Reply 6: Thanks for this suggestion, but we believe the literature review of methods for deriving vulnerability factors is best placed as a step of our process in the methodology section. The idea of our reproducible methodology is that for each community setting and/or hazard a review of the literature has to be done to select appropriate variables, as the factors may change depending on the hazards focused in the particular application of the method. Each step in the defined method is shown in Figure 4.

7. Overall, the maps constructed are good. But to me the Methods section is too long (which is ok). I would suggest remodeling the paper with a focus on the methods proposition rather than studying with a very good methodological section. This will give

validity to the long method section. (10.1016/j.apgeog.2011.07.002) A good paper which focus on methods first and then its application.

Reply 7: Following your recommendations we rearrange the methods in order to separate them from the results (now in a section 3 with the resulting maps). But our approach relies on the co-construction of the methodology with the community, therefore much of the used methods are presented as a result. We now include in section 2 a note for those with the intention of replicating the method:

"The methodology for assessing the physical vulnerability of a community based on physical vulnerability indicators and community-generated data is formed of three stages (Figure 4), namely: Project scoping, Data collection, and Data validation and modeling. In the following sections we will present the result of co-constructing and iterating the methodology with the community researchers. Our advise for those who will replicate this process is to assess the particular needs of each community and reinterpret the approach we are showing next."

8. More details on data analysis on qualitative assessment from community is needed. "e present the positive outcomes from the application of our methodological approach, this evidence was drawn from informal conversations with the community leaders and researchers and, is presented in quotes". How was this evidence analyzed? Thematic analysis, etc?

Reply 8: In the final step of methods section "Data Validation and Modeling" there is an action named: Presentation of maps to the community, presented in section 3.3.4 that included follow-up sessions to evaluate the use of the final maps. The quotes are taken from interviews with the residents of the community months after they had the physical vulnerability maps during a follow-up session. Therefore this is evidence of how they have used the information as a tool for decision making in the neighborhood, we also included a new figure showing the actions taken by the community for mitigation and adaptation.

9. Some paras are too small. Merge them.

Reply 9: Thanks for the suggestions, we have rearranged some portions of the manuscript hoping to resolve this kind of issue.

10. Some references are incomplete. Recheck.

Reply 10: We checked and updated all references lacking doi and url.

Overall, a well-written and excellent paper.

Reply: Thanks to your comments and recommendations, we have greatly improved the manuscript.

- 1. Does the paper address relevant scientific and/or technical questions within the scope of NHESS?
- 2. Does the paper present new data and/or novel concepts, ideas, tools, methods or results?

This approach is not new, but it is rarely used in public policy. Contrary to what the authors claim, there are numerous empirical examples of community risk management, but they are rarely published, and especially not very systematic. The presentation of this methodology is an important contribution, although the authors' advice now would be to generalize this type of experience through comparative studies. The paragraph "Disseminating and upscaling risk reduction practices" points in this direction; it is important to compare experiences and generalize them to regional or national policy levels.

Furthermore, it is always worth pointing out to the authors that vulnerability is not necessarily greater in countries in the Global South, which have "precarious infrastructures." It is primarily the lack of information systems and governance that is the problem, in both developed and developing countries. Recent events in Germany, Valencia, Spain, and Texas demonstrate this.

Reply 1: Thanks for this comment, we acknowledge that climate variability and climate change are pressing issues around the world. Our emphasis results from our perspective, given that this is a Global South study case, and given that is merely a reflection of how our communities, contrary to those of the developed world, have limited access to governmental support, financial insurances and other mechanisms to foster, mitigation, adaptation or even recuperation. Therefore knowledge about physical vulnerability becomes a tool for them to adapt.

Line 30: The lack of a detailed map is not a data problem; it's a governance problem. Therefore, the question is whether researchers should fill the data gap, or whether they should participate in improving governance at all levels!

Reply 2: We agree with your statement, this is now updated in the manuscript.

"The effort is hampered by governance gaps regarding hazard susceptibility and physical vulnerability of these urban neighborhoods which limits the establishment of effective disaster risk reduction (Sachs et al., 2021)."

1. Are these up to international standards?

Not concern

**Reply 3: Thanks.**

1. Are the scientific methods and assumptions valid and outlined clearly?

The method is clear and relevant. The hypotheses are based on participatory approaches, which simultaneously address information gaps, prepare populations for

adaptation measures, and, above all, enable a learning process and the construction of a memory among the population.

Participatory mapping is not new, and there is a vast amount of literature on the subject. The interest of the article therefore lies not in the mapping methodology, but in the results obtained and their interpretation and appropriation by community members.

**Reply 4: Thanks.**

1. Are the results sufficient to support the interpretations and the conclusions?

The results are numerous and well explained.

**Reply 5: Thanks.**

1. Does the author reach substantial conclusions?

The discussion section is overly detailed. It would have been interesting to divide it into two sections: a genuine discussion section and a section explaining the consequences of the approach and the application of the methodology to adaptation measures. For example, the article mentions houses that have been demolished; this is an interesting consequence, but how many? And where? More details on the concrete results of the work and the consequences for adaptation actions would be welcome.

Reply 6: We changed the discussion section and now includes: 4 Discussion: including the discussion of the study; 4.1 Applications: including the responses and actions from the community after they have been using the results of the study and, 4.2 Limitations: maintaining the original content. We also included a new map with the actions from the community with a specific location.

Similarly, how can this methodology be integrated into general policies or urban or territorial planning methodologies?

Reply 7: We complemented the conclusions section with a comment on how the methodology can be integrated in the territorial planning, primarily in the Colombian context:

In the Colombian context, Ley 1523 de 2012 already mandates the integration of citizen-generated data into disaster management plans at different territorial scales. However, this requirement is rarely enforced, due to both a lack of incentives for producing such data and the absence of effective mechanisms for incorporating it into official plans. The proposed methodology can help unlock this potential by providing a practical framework for doing so. Beyond Colombia, the approach can be adapted to strengthen academic—community collaboration and to generate more granular data that grassroots organizations can use at neighborhood or local scales, particularly when institutional data is unavailable.

1. Is the description of the data used, the methods used, the experiments and calculations made, and the results obtained sufficiently complete and accurate to allow their reproduction by fellow scientists (traceability of results)?

This experiment is well described, although there are some uncertainties about the parameters used and the populations involved:

- The indicators are based on physical measurements of the buildings. Wouldn't it be beneficial to also include more social characteristics? For example, education levels, economic status, perception indicators, etc.

Reply 8: Although we included social variables in this exercise as part of the household survey, they were intended for community governance, and along with the community leaders it was decided that the priority was to conduct a physical vulnerability assessment in the neighborhood.

- The article mentions "community leaders," "community researchers," "association," "16 senior members," residents, etc. There is some imprecision at this level. Would it be possible to specify who participated, at what level, and what the characteristics of these participants were?

**Reply 9: We clarified this throughout the manuscript.**

There is some doubt about the validity of the maps, as indicated in Chapter 4.0.1, Limitations.

Reply 10: There is no doubt that this was a thorough process that involved iteration, revision, and discussion. We highlight some limitations for those who want to replicate this methodology in the future. The maps have proven to be of great value for the community, as mentioned in the conclusions section.

- In the maps in Figure 8, it would be important to locate the living space of the 16 senior members. Their living space can indeed influence their perceptions, as we see clear differences between the groups. - Similarly, in 3.3.3, it would be important to specify who was present during the presentation. The article mentions a presentation to the community, how many participants there were, in relation to the number of residents?

Reply 11: Figure 6 presents the division of the neighborhood made to facilitate data collection, one community researcher was responsible for each area. This approach allowed the data gathering process to be more focused and efficient, leveraging the trust that residents had in local researchers. Said trust was particularly important, as the community was generally hesitant to share information due to the pandemic and the recent influx of migrants it had prompted. For the final presentation of the maps, the community invited residents from all sectors, totaling around 16 participants, including children, adults, and elderly residents. We have updated this in the section 3.3.4 Presentation of maps to the community.

- The research effort is also not specified. How many researchers? How long did it take to complete the entire process? What was the budget? Given this data, is the experiment truly reproducible?

Reply 12: We included some context on the methods section and the link to the URBE-Latam project for further information about the scope and funding:

"The project team brought together community researchers with long-standing experience collaborating with academic partners, researchers from local universities, geologists from the British Geological Survey, and researchers from the University of Glasgow. This diverse and interdisciplinary collaboration was developed within the framework of the URBE-Latam project, funded by UKRI (https://gtr.ukri.org/projects?ref=ES\%2FT003294\%2F1), highlighting the integration of local knowledge and scientific expertise."

1. Does the title clearly and unambiguously reflect the contents of the paper? No problem.

**Reply 13: Thanks.**

1. Does the abstract provide a concise, complete and unambiguous summary of the work done and the results obtained?

Very good summary.

**Reply 14: Thanks.**

1. Are the title and the abstract pertinent, and easy to understand to a wide and diversified audience?

No problem.

**Reply 15: Thanks.**

1. Are mathematical formulae, symbols, abbreviations and units correctly defined and used? If the formulae, symbols or abbreviations are numerous, are there tables or appendixes listing them?

Little formula, but very understandable.

**Reply 16: Thanks.**

1. Is the size, quality and readability of each figure adequate to the type and quantity of data presented?

Good

**Reply 17: Thanks.**

1. Does the author give proper credit to previous and/or related work, and does he/she indicate clearly his/her own contribution?

Multiple authors and multiple institutions. The article discusses transdisciplinarity. It would be important to clarify the role of each institution or discipline in the work carried out.

**Reply 18: We have included within the text how each institution participated in the process.**

1. Are the number and quality of the references appropriate?

No problem

**Reply 19: Thanks.**

1. Are the references accessible by fellow scientists?

No problem

**Reply 20: Thanks.**

1. Is the overall presentation well structured, clear and easy to understand by a wide and general audience?

The overall structure of the article is very good and easy to understand.

**Reply 21: Thanks.**

1. Is the length of the paper adequate, too long or too short?

Good

**Reply 22: Thanks.**

1. Is there any part of the paper (title, abstract, main text, formulae, symbols, figures and their captions, tables, list of references, appendixes) that needs to be clarified, reduced, added, combined, or eliminated?

All elements are well described

**Reply 23: Thanks.**

1. Is the technical language precise and understandable by fellow scientists?

There is little technical language and it is quite accessible

**Reply 24: Thanks.**

1. Is the English language of good quality, fluent, simple and easy to read and understand by a wide and diversified audience?

Very good

**Reply 25: Thanks.**

1. Is the amount and quality of supplementary material (if any) appropriate?

Not concerned

**Reply 26: Thanks.**

**Conclusion:**

This article is worth publishing because it contributes to the need to work on risk management at the local level. The methodology is very clear and predicts good results.

Minor corrections may be requested, particularly regarding the role of each institution o discipline in transdisciplinarity, the nature and number of participants, and the possible or implemented adaptation measures.

Reply: Thanks to your comments and recommendations, we have greatly improved the manuscript.